# Prevention and Harm Reduction Interventions for Adult Gambling at the Local Level: An Umbrella Review of Empirical Evidence

**DOI:** 10.3390/ijerph18189484

**Published:** 2021-09-08

**Authors:** Veronica Velasco, Paola Scattola, Laura Gavazzeni, Lara Marchesi, Ioana Elena Nita, Gilberto Giudici

**Affiliations:** 1Psychology Department, Milano-Bicocca University, 20126 Milan, Italy; 2Piccolo Principe Social Cooperative, 24061 Bergamo, Italy; paola.scattola@gmail.com (P.S.); lauragavazzeni@piccoloprincipe.org (L.G.); laramarchesi.bg@gmail.com (L.M.); ele.nt@live.it (I.E.N.); gilbertogiudici@piccoloprincipe.org (G.G.)

**Keywords:** gambling, prevention, harm reduction, effectiveness, review, implementation

## Abstract

Concerns about negative consequences of gambling diffusion are increasing. Prevention and harm reduction strategies play a crucial role in reducing gambling supply and harms. This study aims to conduct an umbrella review of the effectiveness of gambling preventive and harm reduction strategies, which can be implemented at a local level and targeted at adults. It was conducted according to the Preferred Reporting Items for Systematic Reviews and Meta-Analyses (PRISMA) statement. Sixteen reviews were analyzed, and 20 strategies were selected and classified in 4 areas with different targets and aims. Reducing the supply of gambling is an effective strategy both for the general population and for risky or problematic gamblers. Demand reduction interventions have been found to have limited effects but most of them are mainly focused on knowledge about risks and odds ratios. Risk reduction strategies aim to reduce contextual risk factors of the area where gambling is provided, change the gambling locations’ features, and modify individual behaviors while gambling. Smoking and alcohol bans or restrictions are considered one of the most effective strategies. Finally, harm reduction strategies targeted at problematic gamblers are potentially effective. Some relevant implementation conditions are identified and the results show inconsistent effects across different targets.

## 1. Introduction

Gambling is conceived as a recreational activity and most people gamble responsibly. However, some gamblers develop problematic gambling behaviors [1] and concerns about negative health, economic, and relational consequences of gambling diffusion are increasing [2].

Regulations have a crucial role in this area. They can facilitate gambling access or, on the contrary, decrease gambling supply, reduce contextual risks, and limit gambling harms. In some countries (e.g., Britain, Germany, or Italy), acts and laws have facilitated the market-led expansion [3,4]. Regulation has contributed to enhancing the offer of gambling and to further its reach into everyday life. Therefore, the social acceptability of gambling has increased, and the risks of this behavior are often underestimated [5,6]. On the other hand, policies can have an important role in preventing risk gambling, reducing risk factors, and minimizing harms related to gambling. For example, cognitive psychology research reveals that people can benefit from policies that require organizations to provide full utilitarian descriptions regarding the tasks and their consequences; specifically, it is argued that such accessibility of information enhances people’s utilitarian (rational) behavior [7,8]. Moreover, several studies showed that the features of the context influence people’s risk preferences and that problem gambling is fueled by contextual factors [9]. At national, regional, and local levels, many preventive policies have been implemented and concerns about their effectiveness are increasing. In the past two decades, many governments have recognized gambling prevention as a public health issue, involving different stakeholders and promoting the “health in all policies” approach [10]. The importance of harm reduction in gambling, consumer protection and responsible gambling strategies have been acknowledged [11]. In some countries (e.g., Italy), efforts to develop both national and local policies have been made; regions and municipalities have committed to defining local policies to reduce gambling supply and risks factors and to involve different local stakeholders (e.g., health services, local police).

The development and implementation of these policies require a rigorous and detailed analysis of their effectiveness. As a response, many studies and reviews about gambling prevention or harm reduction have been published in the last 10 years, but some research gaps still remain. First of all, many reviews try to identify the effectiveness of different policies or actions, but they do not consider the conditions of implementation (e.g., [3,12]). Health and policy interventions are epistemologically and methodologically complex, the context of implementation is multifaceted and dynamic and the conditions of implementation impact the effectiveness of such interventions [13,14]. Strategies can be effective or not depending on the way they are implemented and the context in which they are used. Moreover, the characteristics and the responsibility of stakeholders involved should be considered [15,16]. Many policies can be effective or not also depending on relational issues and the role that each stakeholder is able to play. Their engagement and preventive role can impact the implementation and the effectiveness of policies [17]. These issues should be considered and described. The realistic synthesis approach underlines the importance of “synthesizing evidence and focuses on providing explanations for why interventions may or may not work, in what contexts, how and in what circumstances” [14]. A more detailed analysis of implementation conditions can provide helpful indications for explaining the absence of significant effects highlighted by previous reviews, and identify the preconditions or circumstances necessary to enhance the effectiveness of each strategy. Moreover, the target of these strategies should be considered more carefully. Preventive and harm reduction strategies can target the general population, gamblers, or risky and problematic gamblers. These three target groups should be considered when analyzing the effectiveness of each strategy.

The second limitation concerns the implementation level. Previous reviews have tended to combine actions that can be pursued by international, national, and local administrations, and businesses. This makes the results of these reviews problematic to apply and use as a guide in policy development and evaluation. In particular, the local level is often overlooked. Most of the strategies proposed can be implemented only by the gambling industry (e.g., pop-up messages and feedback, limits on bets or note acceptors) or at the national level (e.g., types of gambling allowed). The actions and conditions of implementation that may be realized by regions, municipalities, or local organizations have been less investigated. A focus on the effectiveness of local policies’ is important for several reasons. The health and social impacts of problematic gambling are often perceived by local communities. Consequently, local policy makers are often more inclined to implement prevention or harm reduction policies compared to national policy makers. This is particularly relevant in countries where national laws have facilitated market-led expansion [18,19,20]. Moreover, local actions allow for the differentiation of policies according to vulnerability levels and the implementation of specific strategies in at-risk areas [19]. At the local level, it is also more feasible to involve stakeholders and community members in policy development and implementation [19,20]. Therefore, different interventions or strategies can be used at the local level than those at the national level. Finally, it is important that both national and local prevention policies follow effectiveness and evidence-based criteria. Otherwise, inconsistent strategies can be suggested and conflicts between different systems may emerge [18,21].

This study aims to conduct an umbrella review of gambling preventive and harm reduction strategies, which can be implemented at a local level (e.g., realized by regions, municipalities, or local organizations) and targeted at adults. Specifically, we want to (1) synthesize the evidence of the effectiveness of these strategies and (2) identify the effective implementation conditions and targets.

## 2. Materials and Methods

The umbrella review was conducted according to the Preferred Reporting Items for Systematic Reviews and Meta-Analyses (PRISMA) statement [22].

An umbrella review compiles evidence from multiple reviews into one accessible and usable document. It focuses on broad conditions or problems for which there are competing interventions and highlights reviews that address these interventions and their results. The final aim is to summarize what is known and what remains unknown and to give recommendations for practice and future research [23].

### 2.1. Search Strategy

The umbrella review includes reviews and meta-analyses about gambling prevention and harm reduction. We conducted a systematic search on PsychInfo, Scopus and PubMed databases for peer-review review publications using the following keywords: (gambl*) AND ((prevention) OR (harm reduction) OR (responsible gambling) OR (social harm) OR (harm minimization)) AND ((review) OR (meta analy*) OR (meta-analyis) OR (Systematic Review) OR (metaanaly*) OR (research synthesis)). Additional reviews were added based on the reference lists of selected papers. The authors also consulted the websites of official bodies (e.g., European Monitoring Centre of Drugs and Drug Addiction (EMCDDA) or National Anti-Drug Department) and three experts on gambling prevention to identify other reviews.

### 2.2. Inclusion and Exclusion Criteria

The search strategy targeted the period from 2000 to the 20 April 2020. Publications had to be peer-reviewed and written in English. More specific inclusion and exclusion criteria have been defined according to the review aim:Population: Only reviews regarding the adult population were included;Intervention: Reviews about the effectiveness of actions, interventions, and policies to prevent or reduce harms and risks of gambling were included. Reviews about risk and protective factors or about gambling disorders treatment were excluded. Only interventions deemed feasible at local level, that can be implemented in specific areas by municipalities, regions, or other local agencies were considered. Reviews about actions which require the involvement of gambling industry or national laws or agreements were excluded;Outcomes: Changes in gambling behavior or related harm were considered as primary outcomes; secondary outcomes pertain to gambling attitudes, perceptions, and intentions. Recollections, acceptance of intervention, or knowledge were not considered;Comparison: Only systematic reviews and meta-analyses were included. Three mandatory criteria of the Database of Abstracts of Reviews of Effects (DARE) had to be met: the definition of a review question, the inclusion of a search strategy, and the presence of some data synthesis [3,24,25]. Reviews that considered randomized and nonrandomized trials and qualitative studies were included to better identify effective conditions of implementation.

### 2.3. Study Selection and Data Extraction

Studies were selected in two steps applying eligibility criteria. Both steps were conducted by two independent reviewers. The first selection was based on title and abstract, and the full text articles were retrieved if either or both reviewers considered a study to be potentially eligible. The second selection was based on full-text articles. In case of disagreement (2 cases), the project group discussed the articles and criteria compliance; the decision to include the article was based on consensus reached.

The following data were extracted based on reviews’ information: review details, intervention, search strategy, inclusion/exclusion criteria, number of studies in the review, classifications, countries analyzed, the results obtained, the authors’ conclusions, recommendations, and funding sources. Data were extracted and the narrative synthetized by three authors and discussed and revised by another two authors. The group discussed the data and results were based on the consensus reached.

A quality assessment procedure (e.g., the Measurement Tool to Assess Systematic Reviews (AMSTAR 2) [26]) was not used in this review, as most of the studies would have been rated of weak quality and eliminated.

Strategies emerging from the papers reviewed have been classified into four main areas according to their aim and target:Supply reduction: strategies aiming to reduce the supply and availability of gambling opportunities for the general population;Demand reduction: strategies aiming to reduce the desire to gamble and prevent or reduce initiation of problematic gambling;Risk reduction: strategies aiming to reduce the risk factors related to gambling. The target of these actions are gamblers, with the goal of decreasing the probability of developing risky or problematic gambling;Harm reduction: strategies targeting risky or problematic gamblers aiming to identify problematic situations and foster the relationship between them and specific health services.

This classification was based on the policy areas of the European Action Plan on Drugs [27], which can also be applied to gambling prevention. A similar conceptual framework was suggested by McMahon and colleagues [3], but we decided consensually to distinguish between risk and harm reduction because these strategy areas have different targets.

The systematic reviews were narratively summarized in accordance with the above conceptual framework.

## 3. Results

The flow of studies through the review process is shown in Figure 1.

After the two steps of selection, 16 reviews were identified. Table 1 summarizes the characteristics of each review.

The quality of reviews diverges greatly, and so does the level of analysis. In particular, the reviews published from 2015 onwards follow more specific methodological criteria and quality assessment. Some reviews are more focused on the significance of the effects; others better consider the conditions of implementation. In addition, the typology of primary studies considered by the reviews is very different: some authors selected only effectiveness studies, some others also included laboratory studies or population survey studies. A narrative synthesis of the empirical research is presented below.

After data extraction and analysis, 20 strategies have been identified. Some strategies have been considered by several reviews, others by just one. The number of primary studies also differs across strategies. Table 2 summarizes the findings for each strategy and reports recommendations for future researches. The number of primary studies considered by each review and the number of unique studies across reviews are reported.

### 3.1. Supply Reduction Strategies

#### 3.1.1. Restricting Gambling Venues and Licenses

The most important strategy to reducing gambling supply is the restriction of the number of gambling venues and criteria for licenses. Reduction in supply showed a decrease in participation, number of frequent gamblers, demand for treatment and number of problem gamblers [3,33,34,40].

Several countries have a capped number of licenses for casinos or gambling houses; however, these limitations often last for a short time, and the quality of empirical research in this area should improve. Long-term policies should be implemented.

This action needs to be considered not just for land-based but also for online gambling. Literature has mostly focused on ways to limit illegal online gambling [40] and licenses or type of gambling allowed [34]. There is no research on local actions such as, for example, limiting public Wi-Fi connection. This area should be investigated, also considering ethical and political repercussions.

#### 3.1.2. Pricing and Taxation

Two reviews presented pricing and taxation as an effective strategy to reduce the gambling supply [34,40]. However, increasing the price of participating in the legal market may increase the attractiveness of illegal markets. Therefore, illegal markets need to be under control for a tax increase to be effective [34].

#### 3.1.3. Limiting Gambling Venue Hours of Operation

Restriction of gambling venues’ opening hours appears to have an impact on reducing gambling-related harms, although some results are inconsistent [3,33,34,40]. The consistency of opening hours across sites and the compliance with the regulations within the local context are fundamental [40]. The conditions of implementation can explain seemingly inconsistent results. A study, for example, showed how the restriction of opening hours is inefficient, but this was partly due to the fact that clubs varied the time of their shutdown hours, and that other places were exempt from the shutdown [42].

Venues’ opening hours’ restrictions also aim to reduce some risk factors related to night hours, such as concurrent alcohol consumption and the opportunity to gamble continuously. Indeed, many studies show problem gamblers preferring to gamble overnight [34].

Additional research is needed to identify the most effective length of time and time of day for shutdowns [33].

#### 3.1.4. Legal Age

Prohibition of youth gambling seems successful in reducing gambling problems and requires adult involvement [34,40]. Preventing the young from coming into contact with gambling by discouraging parents from giving scratch or lottery tickets as gifts to children, appears to be an efficient strategy to reduce gambling harm. Some researchers showed a link between parental facilitation of gambling and increased gambling behaviors, positive attitudes about gambling and risky gambling among adolescents [43]. However, the implementation of this age limit is problematic. Enforcement of the legal age can be obtained by increasing inspections of gambling venues, enacting penalties [34], and by sensitizing families and parents [40].

#### 3.1.5. Limiting Accessibility to Gambling Venues

Placing gambling venues away from people and limiting their accessibility is a controversial strategy in literature, but it is considered potentially effective. Some authors suggest that the efficacy of these actions is susceptible to contextual variations, such as demographic profile, socio-economic characteristics, and other risky behaviors’ availability [34,40,41]. However, there is some evidence that shows that gambling harm may be higher in locations closer to gambling venues and that distance from venues matters more than gambler preferences about the kind of location.

The influence of distance is also determined by the interaction with other factors. Young and Tyler [41], for example, consider time and distance availability, involvement, and interaction with customers.

### 3.2. Demand Reduction Strategies

#### 3.2.1. Restricting Advertising

The majority of countries have a consumer protection legislation that requires “truth in advertising,” which should theoretically prevent attempts to portray gambling as harmless, safe, or a good way to make money. Beyond this, many countries have additional constraints on gambling advertising [40].

The actual impact of advertising on consumer behavior is complex to understand, and little is known about the effects of gambling advertising on gambling behavior [40]. However, many studies have proved that gambling advertisements have a great impact on the propensity to gamble among problem gamblers. It is reasonable to hypothesize that advertising contributes to a positive attitude about gambling and an increase in engagement when it is offered [44]. Moreover, the proliferation of commercial advertising and gambling opportunities has further increased its social acceptability [6].

#### 3.2.2. Information/Awareness Campaigns

Information campaigns seem to raise awareness of the role of probability laws and skills in gambling, avoiding gambling fallacies. However, they are not associated with any decreases in actual gambling behavior [38,40]. Awareness initiatives appear to have a very limited positive impact if people are not explicitly asked to attend to the information or have no intrinsic interest in it [40]. Irrational beliefs about gambling are highly idiosyncratic and context-bound, difficult to prove false, and stem more from the selective misuse of information than from a lack of knowledge about gambling [38]. Gainsbury et al. [34] suggest the importance of more specific campaigns targeted at parents to increase awareness of the importance of restricting youth gambling.

Reviews showed that there is no evidence of effectiveness of venue signage [37].

#### 3.2.3. Educational Interventions

Educational interventions usually intend to change knowledge, attitudes, beliefs, and skills, which are seen as critical factors that influence both the decision to gamble and the progression to problem gambling [40]. Research has mainly focused on youth prevention program evaluation, and few studies regard adult educational interventions [12]. Most adult educational interventions focus on knowledge, misconceptions and fallacies, but they have little impact on behaviors [38]. In some counties, “Responsible Gambling Information Centers” (RGICs) are located within gambling venues to provide information and education about the risks of gambling and to identify and support visitors who are experiencing problems with gambling. An evaluation of their effectiveness showed that visitors appeared to modify misconceptions about randomness but did not have any immediate or long-term impact on gambling behavior [40].

Most of the studies focus on the personalized normative feedback (PNF) or personalized feedback intervention (PFI) approach, which aim to change behaviors by highlighting erroneous beliefs and facilitating a discrepancy between perceived and actual norms [3,5,12]. This kind of intervention is considered a potentially effective, low-cost, and easily disseminated strategy for reducing at-risk gambling as a harm reduction preventive strategy. However, the implementation should be cautious. Grande-Gosende et al. [5] have warned about the use of this strategy because it may cause a “boomerang effect” when targeting low-frequency gamblers.

More specific educational interventions for adult gamblers seem to be successful in reducing gambling behaviors. These programs aim to develop participants’ skills to cope with gambling, change gambling attitudes, and restructure cognitive processes [38]. Moreover, they are delivered by professionals with a close relationship with participants (e.g., medical professionals). Other educational approaches can be developed to train people on how and when their gambling behaviors might indicate risk. These interventions may also be useful in designing targeted and individualized interventions for problematic gamblers [30].

A specific area for educational intervention is family/parent training interventions [34,40]. Interventions that strengthen families and create effective parenting practices are considered one of the most powerful ways to reduce adolescent problem behaviors and reduce problems at later ages. However, more studies are needed in this area because the certainty of evidence is very low [12].

### 3.3. Risk Reduction Strategies

#### 3.3.1. Restricting Access to Cash

The restriction of ready access to cash can be considered as a moderately effective strategy [3,33,37,40]. A study found that the removal of ATMs in the vicinity of gambling venues reduced the expenditure on electronic gambling machines overall by 7% [45], and other authors identified this strategy as potentially effective. Moreover, problem gamblers access cash machines more frequently than regular gamblers.

However, reviews identify a lack of empirical research examining the effectiveness of monetary restrictions. Moreover, literature is mainly focused on gambling venues’ proximity to or removal of ATMs, while consequences of those actions can also be valid for other places that allow access to money, such as cash for gold stores or pawn shops.

#### 3.3.2. Placing Gambling Venues Away from Vulnerable Populations

Particular social contexts and subpopulations will be most vulnerable to increased levels of exposure. The effectiveness of reducing the number of gambling venues and placing them away from vulnerable populations reducing risk factors has good empirical support [40,41]. Young people, people with other addictions, and people with low socio-economic status are identified as vulnerable populations in these studies. A relationship between social disadvantage, proximity of gambling venues, and gambling harm has been highlighted and a positive correlation between electronic gaming machine (EGM) concentration and social disadvantage areas has been identified in many countries [41].

#### 3.3.3. Ambient Natural Lighting

Specific gambling location design elements may be related to continued play, such as the lack of windows and low ambient lighting. The lack of lightning and other design elements seems to promote gambling behavior mainly among current gamblers; a plausible mechanism might be their prior conditioned association with gambling [40]. However, this issue requires additional research.

#### 3.3.4. Clocks and Time Awareness

Becoming more aware of time spent on a gaming session can have a positive influence on gambling behaviors and promote responsible gambling [46]. In order to increase time awareness while gambling, researchers have studied the value of clock use in gambling venues [31,32,33,40].

Most of the studies are focused on on-screen clocks. They were associated with improvements in keeping track of time and staying within desired time limits, but they had no effect on reducing session length or expenditure. Moreover, studies report that only a small portion of people use them and consider clocks helpful.

Few studies focus on room clocks. Also in this case, most gamblers did not refer to a clock while playing, and only a few patrons chose to activate the optional alarm clock feature.

Other strategies to facilitate time awareness focused on promoting a “caring” relationship between manager/employee and gambler. Although potentially beneficial, these have not been considered in reviews and evaluation studies.

#### 3.3.5. Machine Location

Two reviews presented machine location as a strategy to reduce gambling risks. There is conflicting evidence about this strategy but, overall, reviews suggest this is an efficient way to counteract gambling [30,40].

Some studies showed that the centrality of the machines’ location has high effects on improving gambling. On the contrary, many other studies report that players who were isolated from others were more likely to play excessively and that players reduced time played and bet sizes when they were observed versus when they were not.

#### 3.3.6. Smoking Bans/Restrictions

The literature investigated the efficacy of smoking bans in gambling venues. Williams et al. [40] consider the restriction of the consumption of tobacco while gambling as one of the most effective strategies; other reviews confirm this conclusion [3,33]. Even if few gamblers perceive a change in their behavior, smoking bans have shown that they influence gambling expenses. For example, in the state of Victoria, the introduction of smoking bans led to an immediate reduction of gaming revenue, indirectly reducing gambling expenses [47]. Requiring people to move from gambling areas to designated smoking areas provides a natural gaming interruption and reduces gambling expenses indirectly [48].

#### 3.3.7. Alcohol Bans/Restrictions

Gambling and drinking often concur, particularly for problem gamblers. A positive correlation between increased drinking and more serious gambling problems has also been demonstrated (e.g., length of play, rate of double-up betting, a play of losing hands, and financial loss). Moreover, alcohol has a disinhibiting effect on gambling restraint and increases risk-taking. Given this knowledge, Williams et al. [40] consider restrictions on alcohol beverages administration while gambling as having significant potential as a harm minimization strategy for problem gambling. It may also be assumed that a ban on the sale of alcoholic beverages and on drinking while gambling, such as smoking bans, can cause gamblers to interrupt gambling sessions and so reduce gambling-related harm. Surprisingly, no other review takes this strategy into account.

### 3.4. Harm Reduction Strategies

#### 3.4.1. Gambling Venues Employee Training

A strategy used in multiple countries consists of training gambling venue employees [28,30,31,34,38,40]. These training activities are aimed at improving knowledge of legal obligations, understanding gambling behaviors, changing attitudes and beliefs about gambling, improving employees’ ability to intercept problem gamblers and to refer them to the competent care service [28,40]. This kind of activity is often considered an early intervention with problematic gamblers. However, it should be included among risk reduction interventions, given that its main aims are to provide a safe gambling environment and to encourage responsible gambling [28,38]. Employees may transmit warning messages and prompt reflection about clients’ gambling behaviors in a more specific and empathetic way than machines. The relationship is particularly important for older gamblers who value friendly staff and a sense of belonging [30]. However, the main focus of training is still on problematic gamblers, who display reliable behavioral cues in gambling venues (i.e., anger, repeated withdrawals from ATMs, etc.).

Evaluation studies show that staff training programs are effective in changing staff members’ knowledge, attitudes, and self-confidence. They increase empathy, value their role in responsible gambling and improve the ability to detect gambling problems. However, they fail in promoting a proactive strategy and in facilitating intervention with some gamblers. Staff display a high level of confusion, doubts, and worries about dealing with problem gamblers [31] and need more specific behavioral skills [28]. Some training interventions have a short-term effect if not supported by the different stakeholders involved [38].

Few studies evaluated the effects on venue gamblers and there is insufficient evidence about gamblers’ outcomes [28]. Future evaluations of staff training should be realized, including more specific information about training components and using rigorous methodological designs [28].

#### 3.4.2. Tests and Screening

Policy measures regarding minimal interventions (screening and brief interventions) to control gambling and related harm are most likely effective [34]. However, many barriers intervene: lack of time, skills, motivation, and organizational factors. Policies and guidelines should promote the use of these interventions and training should support professionals. These interventions are more effective when combined with specific training, such as training targeted at gambling venue employees (see previous section), general practitioners [34], or nurses of geriatric patients [30]. Otherwise, staff and professionals may have difficulties in identifying indicators of problematic gamblers, and most of all may be reluctant to intervene [37].

#### 3.4.3. Helplines and Information about Care Services

In all countries, many efforts are made to promote helplines and health services to support and treat gambling disorders. Literature shows how machine and instant lottery gamblers are more aware of gambling problem reduction initiatives, and this suggests that they notice helplines and information cards on slot machines [40]. A study evaluation of an informational campaign that used radio, newspaper and billboard advertisements reported a 70% increase in calls to the helpline and a 118% increase in customers requesting treatment [49]. However, Livingstone et al. [37] conclude that there is no evidence of signage in gambling venues’ effectiveness. More studies are needed in this area to better understand the real use of this information.

#### 3.4.4. Precommitment

A deeply studied harm minimization technique is precommitment [3,30,31,37,39,40]. Precommitment involves strategies to limit one’s gambling, such as money limits or time limits. A conclusive statement on the effectiveness of this cannot yet be offered due to methodological problems, implementation discrepancy, and inconsistent results. However, the reviews analyzed offer some suggestions about the effectiveness of implementation conditions [40]. First of all, it is important to distinguish the target. Precommitment can be effective for some persons and increase problems for others [31,39]. Matheson et al. [30] consider precommitment as a strategy to enhance people’s ability to control their gambling in the early stages of problematic gambling but also as a way to motivate problematic gamblers to seek help. McMahon et al. [3] warn about unintended consequences with problematic gamblers increasing gambling expenditure: Problematic gamblers are more likely to both set higher limits and exceed these limits. Second, specific characteristics may influence effectiveness. Livingstone et al. [37] underline the importance of a universal and binding precommitment system. Matheson et al. [30] suggest focusing on limiting the time spent because it is one of the most important indicators of problematic gambling.

#### 3.4.5. Self-Exclusion

Self-exclusion is an extreme form of precommitment: Gamblers who believe that they have a problem from gambling can voluntarily ban themselves from entering one or more gambling venues [36]. As with precommitment, a conclusive statement on its effectiveness cannot yet be offered because of methodological issues, implementation discrepancy, lack of evidences about long-term effects and unknown causal relationships [3,29,30,31,32,36,37,38]. The main inconsistent results pertain to gambling behaviors. Some reviews conclude that many self-excluders continue gambling inside or outside the excluded venues [29] or that they comply with exclusion during the defined period, but changes are not maintained once excluders returned to gambling [3,32]. Other reviews are supportive of the effectiveness of this strategy [31,36]. All reviews agree that self-excluders generally experience benefits from the program. Self-excluded gamblers reported a reduction of negative experiences (e.g., depressive symptoms) [32] and improvements in multiple individual and social areas: self-confidence [32]; quality of life [30,32]; psychosocial functioning [3]; family relations [3]; and work performance [3,32]. Self-exclusion also has the potential to provide additional external constraints when motivation falters. However, this function is conditional on some specific factors [40]: irrevocability of bans or availability of longer ban lengths; extent of application of the ban; perceived and actual chances of re-entry detection; consequences of detection; consequences of failing to detect; and availability of complementary treatment. Based on available evidence, Gainsbury [36] and Kotter et al. [29] identified some elements that should be included in self-exclusion programs: promotion and clear information about the program; early detection of problem gambling by venue staff; easy access to the program; minimum of 6-month periods for exclusion; inclusion of all gambling segments; venue access controls; no incentives during the self-exclusion period; information about educational and treatment resources; active steps to identify and remove self-excluded persons by operators; reinstatement process; training programs for operators; program evaluation.

## 4. Discussion

The current umbrella review examined the effectiveness of gambling preventive and harm reduction strategies, which can be implemented at a local level and targeted at adults. Sixteen reviews were analyzed, and 20 strategies were identified and classified into 4 areas: supply reduction, demand reduction, risk reduction, and harm reduction.

Reducing the supply of gambling seems to be an effective strategy both for general populations and for risky or problematic gamblers. Regarding demand reduction strategies, restricting gambling advertising has been reported to be a promising strategy. Conversely, social campaigns and educational interventions seem to not be effective, although this result depends on the fact that most of them aim to change only the knowledge and misconceptions about gambling. More specific interventions focused on skills, relations, and attitudes appear more promising. Risk reduction strategies can be divided into three groups: actions aimed at reducing contextual risk factors of the area where gambling is provided (e.g., restricting access to cash); actions related to changes in the features of gambling locations (e.g., ambient natural lighting); and actions aiming to change individual behaviors while gambling (e.g., smoking bans/restrictions). There is conflicting evidence about interventions aimed at changing the features of gambling locations. Smoking and alcohol bans or restrictions are considered one of the most effective strategies. Finally, harm reduction strategies targeted at problematic gamblers appear potentially effective.

This review also aimed to identify effective implementation conditions and targets. Health and policy strategies consist of complex interventions influenced by a multifaced context and dynamic conditions [13,14]. The effectiveness of these strategies depends on the way in which they are implemented, on the context in which they are used and on the target they reach. The published umbrella review [3] did not consider these elements. In contrast, our review identified some relevant conditions that can explain some of the inconsistent results reported in the previous reviews. Implementation monitoring and fidelity are critical for supply reduction strategies or contextual risk factor reduction. For example, the prohibition of youth gambling has been theoretically successful, but compliance is challenging and additional control and sensibilization strategies have become necessary to guarantee its effectiveness. Another relevant condition is implementation consistency. For example, the limit on operating hours is effective only if it is homogeneous and implemented in a wide area and with high compliance. It is also important to ensure that the policy is sustained in the long term; otherwise, positive effects will be difficult to assess and maintain. The engagement of stakeholders and community members has also been shown to be relevant for several strategies, such as pricing and taxation, educational interventions, the training of gambling venue employees, and precommitment and self-exclusion. However, many barriers intervene in the implementation of these strategies. For example, test and screening use is reduced because of a lack of time, skills, motivation, and organizational limits. The intersectoral and community-based approach [19,50] and the responsible gambling perspective [11] should be reinforced, promoting collaboration among different agencies and encouraging a “caring” relationship between community members and gamblers. For example, some strategies may be reinforced through an active role played by gambling venue staff in interactions with customers. The results show that it is necessary to sustain strategy implementation with forms of assistance and capacity building to overcome barriers to action. For example, test and screening use are more effective when combined with specific training. More efforts are needed improve stakeholders’ skills, and reinforce collaborations between them and health services. Finally, some strategies should respect specific characteristics to assure target compliance. For example, reviews of precommitment and self-exclusion programs identified some relevant characteristics related to obligations, failure consequences and limitations. Moreover, the training of gambling venue employees should reinforce behavioral skills.

The results of this umbrella review also showed the relevance of considering the targets when reviewing evidence. For example, the PNF or PFI may cause a “boomerang effect” when targeting low-frequency gamblers. Precommitment can be effective for some persons and can be problematic for others and self-exclusion programs should focus on the early detection of problematic gamblers. Moreover, educational programs are more effective when intermediate targets, such as professionals with a close relationship with participants, are considered. Social campaigns are considered more effective when targeted at parents. The training of gambling venue employees is more effective when stakeholders are involved.

Another peculiarity of this review is its focus on prevention and harm reduction strategies that can be implemented at the local level, for example by municipalities, regions, or other local agencies. The local level may have several potentials. First, local agencies may have more control and pay attention to the implementation conditions. They also have more opportunities to focus on specific targets or on particular at-risk areas than national agencies do. Both these aspects were shown to be relevant by the results of this review. At the local level it is also more feasible to activate a multisetting strategy and to involve stakeholder and community members in strategies implementation [19]. On the other side, local policies have some limitations. They struggle in regard to guaranteeing strategy coherence and homogeneity, which were identified as relevant elements for increasing effectiveness (e.g., for the limitations of venue operating hours). Moreover, local policies may be in contrast with national policies, creating confusion and tensions [18,21].

### Limitations

The review presents some limitations related to the methodology used. For an umbrella review to be useful requires the pre-existence of the narrower component reviews and its output is inevitably limited by the content of the included reviews and the level of synthesis provided. Moreover, it does not include statistical processing typical of meta-analyses to assess the results. However, its ability to synthesize the evidence across a breadth of literature offers a good overview and several recommendations [3,23].

First, most of the studies considered were conducted in Canada, Australia, the USA, and the north of Europe, thus offering limited insights into the relevance of findings in societies and countries with different social norms and cultures. Second, not all reviews displayed the same level of methodological quality. The criteria used to select the studies were different across reviews and in some cases the list of primary studies was not available. The sample size and effect size are not reported in most of the reviews. However, consistency of results was verified, and differences in implementation and methods were considered. Finally, although most of the reviews reported conflicts of interests and funding sources, primary studies’ sources were not reported. More specific distinctions should be considered according to this issue.

Other limitations concern the local perspective. The reviews analyzed did not consider the local perspective and no information was given about the level of implementation of the strategies considered. Future research is needed to confirm this evidence at both the national and local levels.

## 5. Conclusions

This umbrella review collected evidence from multiple reviews and combine them into one comprehensive and functional document about the effectiveness and conditions of the implementation of gambling preventive and harm reduction strategies, which can be applied at the local level and targeted at adults. The results contribute to understanding the social and environmental factors that can determine problematic or addictive gambling behaviors and of the effective strategies to prevent them.

Several research and practical implications can be suggested. This review identified some research gaps. First, most of the studied works identified methodological limitations which prevent the reaching of conclusive statements on their effectiveness. Moreover, some strategies have been evaluated by many authors while others have been considered by just few studies. New rigorous studies are needed. Second, effectiveness studies should better consider fidelity and compliance in implementing strategies. Some strategies require verification, their violation should be penalized, and their facilitators should be identified. The third gap regards the role of the actors involved. Most of the strategies are related to relations within a community or between operators and customers. More studies about the role of communities and stakeholders should be conducted to identify facilitators and obstacles in implementations, verify the effectiveness of new strategies, and valuing social capital to prevent gambling and its risks. Moreover, some strategies that have been currently developed and evaluated only by gambling industries could be adapted for implementation at a local level. For example, time management strategies or feedback while people are gambling, or setting gambling limits are now considered only through machines or online platforms. These strategies may also be put into practice by trained staff who will be able to personalize the messages and use personal relationships to improve effectiveness. This approach is even more important to enable warning signs to encourage reflection rather than just offer information about winning odds [30]. In addition, new strategies to limit online gambling through local policies (e.g., limiting public Wi-Fi connection) should be investigated. Furthermore, more studies should be conducted to assess the effectiveness of the strategies reported in this review implemented at a local level and of new strategies that are feasible only at the local level. More studies are needed to evaluate the effectiveness of gambling prevention strategies in other countries and cultures. Finally, more efforts are needed to validate the knowledge from other areas of addiction (tobacco, alcohol, drugs) in the gambling area. Gainsbury [34] already used this approach, examining the evidence-based alcohol policies [35] and giving recommendations about gambling policies.

From a practical point of view, this review offers an overview of the strategies that can be implemented at the local level to prevent and reduce the gambling harms. It also provides suggestions about the implementation conditions that require more attention. Policy makers and practitioners should use this information to consider the pre-conditions necessary to implement strategies, define policies in more detail, and support and monitor strategies application.

## Figures and Tables

**Figure 1 ijerph-18-09484-f001:**
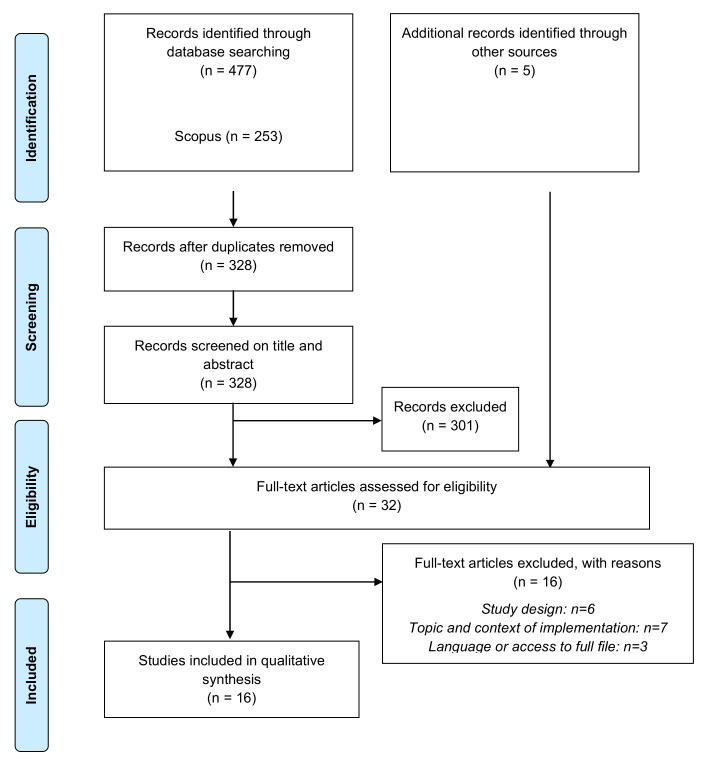
PRISMA flow diagram of studies.

**Table 1 ijerph-18-09484-t001:** Reviews characteristics.

Author	Year	Main Goal	N	Main Findings
Grande-Gosende [5]	2020	To critically assess the existing literature on the effectiveness of prevention programs aimed at reducing the prevalence of gambling problems among young adults and identify the specific preventive components used.	9	Gambling prevention programs mostly followed a selective or indicated prevention strategy. The personalized normative feedback (PNF) approach is the preferred strategy for reducing at-risk or problem gambling among young adults, showing at least a moderate positive effect in most of the included studies. Improving mathematical knowledge based on a gambling framework was shown to increase the ability to calculate gambling odds and resistance to gambling fallacies at the long-term assessment but did not reflect an overall reduction in gambling behavior.
Forsström [12]	2020	To assess the certainty of the evidence relating to different gambling preventive measures in the context of educational programs and consumer protection measures. To present and discuss the shortcomings identified in eligible studies to better understand how preventive measures should be designed and tentatively identify the probable results of future studies.	28	Results indicate a potential effect of PF interventions. For the remaining interventions, the certainty of evidence was very low.
Beckett [28]	2020	To evaluate the current evidence of the impact venue staff training programs in responsible gambling have on venue staff and gamblers.	22	Staff training programs provide some benefit to staff members. Few studies evaluated the effects on venue gamblers and there is insufficient evidence to make a clear causal link between responsible gambling staff training programs and the reduction of gambling harm in customers.
McMahon [3]	2019	To evaluate the systematic review evidence base on the effects of prevention and harm reduction interventions on gambling behaviors, and gambling-related harm. To examine differential effects of interventions across sociodemographic groups.	10	Precommitment and limit setting showed positive findings, but such interventions are limited by the extent to which users adhere to voluntary systems. Potential negative unintended consequences are possible for high-risk and problem gamblers. There was some preliminary support for reduced opening hours of gaming machines; smoking bans; personalized feedback interventions; removal of large note acceptors; maximum bets; and removal of ATMs. The quality of the included systematic reviews was found to be low.
Kotter [29]	2019	To understand who is participating in land-based self-exclusion programs, and the differences between excluders from casinos and those from other land-based gambling venues. To investigate how gambling behavior changes after self-exclusion for abstinence, reduction, increasing, breaching and relocation. To detect the prevalence of gambling problems and symptoms of other mental disorders in self-excluders. To understand whether exclusion reduces prevalence rates, other mental disorders, and improves mental health.	19	The results revealed wide ranges of changes in gambling behavior after exclusion. Gambling-related problems declined after self-exclusion, and several aspects of mental health improved. However, many self-excluders continued gambling inside or outside the excluded venues. Improvements in practice are needed.
Matheson [30]	2018	To summarize the literature and available evidence on the prevention and treatment of PG among older adults. To inform interventions for prevention and treatment and identify literature gaps.	247	Education for seniors should consider cultural differences, comorbidities, stigma associated with help seeking, and family supports. It should include awareness of the potential risks of gambling, self-diagnosis, cognitive distortions, and odds delivered in various formats to accommodate cognitive ability (e.g., dementia). Given that older adults engage in gambling as a social activity, it may be necessary to monitor accessibility to venues and frequency of patronage. Prevention training for the gambling industry should provide information on risk factors specific to older adults. Education in prevention for primary care professionals is imperative to ensure that older adults who access health-care services with gambling concerns are identified quickly. Training of staff at gambling venues, family, primary care staff, and staff working at senior residences is crucial.
Ladouceur [31]	2017	To identify empirically grounded responsible gambling studies to guide evidenced-based effective responsible gambling strategies.	29	Self-exclusion programs demonstrate some effectiveness as a component of RG programs despite various limitations including low utilization rates, breaching the agreement, and minimal evidence about the long-term outcomes. Although there is an increase in research focusing on behavioral indicators of gambling-related problems, the current state of knowledge remains underdeveloped. There is a lack of conclusive evidence about integrating these tools within fully developed RG programs. There is empirical evidence that suggests that limit setting can be effective for promoting RG. However, it is important to remember that limit setting is only effective for some individuals; it can increase gambling problems for others. Venue staff providing assistance to patrons experiencing problem gambling demonstrates partial effectiveness as a useful RG initiative.
Drawson [32]	2017	To collate the empirical evidence to date on the effectiveness of protective behavioral strategies in gambling.	33	Self-exclusion was the only strategy with sufficient evidence to be recommended; however, even the quality of this evidence was not high and requires improvement in future studies before clinical recommendations can be made. The findings on time-limit setting were inconsistent. Setting a monetary limit was much more highly endorsed than setting a time limit. Conclusions about the usefulness of other behavioral strategies for gamblers cannot be made
Tanner [33]	2017	To identify and evaluate industry/environmental-level harm reduction approaches to gambling.	27	Further research is needed to determine effectiveness of mandatory shutdowns to identify the most effective length of time and time of day for shutdowns. Few studies examined the effectiveness of on-screen clocks. Anecdotal results are available. While there is little research into EGM caps, preliminary evidence seems to suggest that it is an ineffective strategy to change gambling behavior. Smoking bans have been found to be effective. Overall gambling expenditure even though few individuals perceive a change in their gambling behavior. Alcohol use and gambling commonly co-occur and research suggests that alcohol may disinhibit gambling behavior but this is an area in need of further research to determine if it may be an effective harm reduction strategy.
Gainsbury [34]	2014	To provide recommendations for international guidelines for harm-minimization policy for gambling including Internet gambling. These recommendations will be based on the framework provided by Babor et al. [35] in relation to evidence-based alcohol policies.	1	Many of the public health policies implemented for substance use may be adaptable to addressing gambling-related harms. The most potential effective policies are legal age limit and regulation of licenses and monopolicies. Other potential effective policies are: price and tax regulation, grief interventions with at-risk and problematic gamblers and opening hours and outlet density reduction.
Gainsbury [36]	2014	To provide a comprehensive understanding of the available evidence to date that is relevant to the establishment and implementation of a self-exclusion program.	14	The assessments of self-exclusion programs internationally generally find that the majority of participants benefit from such schemes. However, the current programs are in need of improvements to improve utilization rates and outcomes over time. A key deficit in current self-exclusion programs is that the majority of problem gamblers do not enter into these agreements.
Livingstone [37]	2014	To review the available evidence for a range of harm minimization practices, particularly those implemented within EGM venues via “codes of practice.”	ns	There is only modest evidence supporting the harm minimization practices. Self-exclusion: there is modest evidence that SE programs are an effective intervention for changing individual (rather than population-wide) gambler behavior and reducing gambling-related harm Signals: there is no evidence of effectiveness. Venue staff screening: there is little evidence of practices where venue staff identify problem gambling behavior and then interact with gamblers so identified. Precommitment: while the evidence base is somewhat limited in demonstrating the effectiveness of universal and binding precommitment systems, it does demonstrate that partial–or optional–systems are not effective population-wide harm reduction strategies. ATM removal: there is modest but reasonable evidence of its effectiveness.
Ariyabuddhiphongs [38]	2013	To review the literature on problem gambling prevention measures, considering both harm reduction and responsible gambling models	70	Problem gambling prevention measures may be classified into the temporal sequence of before, during, and after gambling. The “before prevention measures” aim to correct misconceptions on and change attitudes toward gambling; their success in reducing gambling behaviors seems limited. The “during measures” that involve structural changes to gambling machines and insertion of warning signage appear to yield mixed results. The “after measures” that feature problem gamblers’ voluntary self-exclusion yield limited success; self-excluded gamblers breach the agreement by returning to the venues to gamble.
Ladouceur [39]	2012	To review the effectiveness of voluntary or mandatory precommitment systems for electronic gaming machines.	17	The presence of methodological limitations preclude any conclusive statement on the effectiveness of precommitment systems on gamblers.
Williams [40]	2012	To propose an etiological framework for understanding how problem gambling develops based on the available evidence and drawing from established models of addictive behavior. To comprehensively evaluate the effectiveness of the various initiatives that have been used around the world to prevent problem gambling based on their demonstrated efficacy and/or their similarity to initiatives that are empirically effective in preventing other addictive behavior. To identify current “best practices” for the prevention of problem gambling.	ns	A very large number of different prevention initiatives exist. There are a few initiatives where almost no direct evidence exists concerning their efficacy, many initiatives where some direct evidence exists, and a small number of initiatives where extensive evidence exists. The most commonly adopted prevention measures tend to be among the least effective ones. When potentially more effective initiatives are implemented, they are typically done in such an inconsequential or perfunctory fashion as to virtually ensure lack of impact. Some best practices suggested are: decrease the general availability of gambling; restrict the use of tobacco and alcohol while gambling; restrict access to money while gambling; and impart knowledge, attitudes, and skills to gamblers to inhibit the progression to problem gambling.
Young and Tyler [41]	2008	To review the potential social impacts of changes in the supply structure of gambling opportunities	272	Several structural characteristics of venues affect participation and problem gambling levels. These include distance from markets, type of gambling, number of EGMs, range of nongambling facilities, the structure of catchments, the level of community involvement, and different systems of ownership and control. More attention needs to be directed towards the socio-spatial relationships between venues and their local clienteles.

N = number of studies included; ns = not specified.

**Table 2 ijerph-18-09484-t002:** Summary of findings–strategies.

Strategies	Reviews (Number of Studies)	N	Summary of Findings on		Recommendations
			Effectiveness	Implementation Conditions	
* **Supply reduction strategies** *					
1. Restricting gambling venues and licenses	Gainsbury, 2014; McMahon, 2019 (2); Tanner, 2017 (2); Williams, 2012 (11)	13	This is one of the most important strategies to reduce gambling supply. However, the quality of empirical research in this area should improve. Preliminary evidence suggests that the EGM caps is an ineffective strategy.	The limitations often last for a short time. Long-term policies should be implemented.	To evaluate long-term policies’ implementation.
2. Pricing and taxation	Gainsbury, 2014; Williams, 2012 (1)	1	This is considered an effective strategy to reduce the gambling supply.	Increasing the price of participating in the legal market may increase the attractiveness of illegal markets. Illegal markets need to be under control for a tax increase to be effective.	
3. Limiting gambling venue hours of operation	Gainsbury, 2014; McMahhon, 2019 (4); Tanner, 2017 (4); Williams, 2012 (3)	6	This seems to have an impact on reducing gambling harms and risk factors.	The consistency of opening hours across sites and the compliance with the regulation within the local context are fundamental.	Further research is needed to determine effectiveness of mandatory shutdowns to identify the most effective length of time and time of day for shutdowns.
4. Legal age	Gainsbury, 2014; Williams, 2012 (8)	8	Prohibition of youth gambling seems successful in reducing gambling problems and requires adult involvement. Evidence shows the link between parental facilitation and increased gambling behaviors.	The implementation of this age limit is problematic. More controls and families’ sensibilization strategies are needed.	
5. Limiting accessibility to gambling venues	Gainsbury, 2014; Williams, 2012 (8); Young & Tyler, 2008 (20)	27	This is a controversial strategy in literature, but it is considered potentially effective.	The efficacy of these actions is susceptible to contextual variations and factors related to interactions.	The interaction with other contextual factors should be considered.
* **Demand reduction strategies** *					
6. Restricting advertising	Williams, 2012 (13)	13	It is reasonable to hypothesize that advertising contributes to a positive attitude about gambling, an increase in engagement when it is offered and to social acceptability.		Further research is needed to understand the impact of gambling advertising.
7. Information/awareness campaigns	Ariyabuddhiphongs, 2013 (2); Gainsbury, 2014; Livingstone et al., 2014 (2); Williams, 2012 (8)	12	They seem to raise awareness of the role of probability laws and skills in gambling, avoiding gambling fallacies. However, they are not associated with any decreases in actual gambling behavior. There is no evidence of effectiveness of venue signage. Specific campaigns targeted at parents may be effective in increasing awareness of the importance of restricting youth gambling.	More targeted campaigns should be developed. Parents are a potential target.	
8. Educational interventions	Ariyabuddhiphongs, 2013 (3); Forsström, 2020 (9); Gainsbury, 2014; Grande-Gosende, 2020 (9); McMahon, 2019 (2); Matheson, 2018 (3); Williams, 2012 (21)	38	Most adult educational interventions had little impact on behaviors. Specific programs aimed at developing participants’ skills, change attitudes and restructure cognitive processes seem to be successful. An evaluation of RGICs showed that visitors appeared to modify misconceptions but did not have any impact on gambling behavior. PNF or PFI is considered a potentially effective, low-cost and easily disseminated strategy.	It is important to involve professionals with close relationship with participants. The PNF or PFI implementation should be cautious because it may cause a “boomerang effect” when targeting low-frequency gamblers.	To develop and evaluate new adult educational programs and parent training.
* **Risk reduction strategies** *					
9. Restricting access to cash	Livingstone, 2014 (1); McMahon, 2019 (2); Tanner, 2017 (2); Williams, 2012 (7)	9	ATM removal in the vicinity of gambling venues can be considered as a moderately effective strategy. Reviews identify a lack of empirical research.	Other cash sources are not considered.	Additional research about its effectiveness is needed. Other cash sources should be considered.
10. Placing gambling venues away from vulnerable populations	Williams, 2012 (5); Young and Tyler 2008 (11)	16	This has good empirical support.	A clear definition of vulnerable populations is necessary.	
11. Ambient natural lighting	Williams, 2012 (4)	4	The lack of lighting and other design elements seems to promote gambling behavior mainly among current gamblers.	No information about different levels of light is available.	Additional research is needed.
12. Clocks and time awareness	Drawson, 2017 (2); Ladouceur, 2017 (2); Tanner, 2017 (2); Williams, 2012 (3)	4	Most of the studies are focused on on-screen clocks. Few studies focus on room clocks. It does not seem effective in reducing session length or expenditure.		Other strategies to facilitate time awareness should be investigated because time awareness can have a positive influence.
13. Machine location	Matheson, 2018 (2); Williams, 2012 (4)	5	There is conflicting evidence about this strategy but, overall, reviews suggest this is an efficient way to counteract gambling.		Both visibility and isolation effects should be considered.
14. Smoking bans/restrictions	McMahon, 2019 (2); Tanner, 2017 (2); Williams, 2012 (9)	11	This is considered as one of the most effective strategies.		
15. Alcohol bans/restrictions	Williams, 2012 (10)	10	This has significant potential as a harm minimization strategy.		Additional research about its effectiveness is needed.
* **Harm reduction strategies** *					
16. Gambling venue employee training	Ariyabuddhiphongs, 2013 (3); Beckett, 2020 (22); Gainsbury, 2014; Ladouceur, 2017 (3); Matheson, 2018 (5); Williams, 2012 (12)	31	Staff training programs are effective in changing staff members’ knowledge, attitudes, and self-confidence. However, they fail in promoting a proactive strategy and in facilitating intervention with some gamblers. Some training interventions have a short-term effect if not supported by the different stakeholders involved. There is insufficient evidence about effects in customers.	The main target is problematic gamblers, but all gamblers should be considered to encourage responsible behaviors. Behavioral skills training is necessary. Stakeholders’ involvement improves effectiveness.	More research about design, implementation and evaluation of employee training is needed. Training aimed at developing staff members’ behavioral skills should be designed.
17. Test and screening	Gainsbury, 2014; Livingstone, 2014 (2); Matheson, 2018 (1)	3	This is most likely effective.	Many barriers intervene: lack of time, skills, motivation, and organizational factors. These interventions are more effective when combined with specific training.	Additional research about the implementation is needed.
18. Helplines and care services information	Livingstone, 2014 (2); Williams, 2012 (4)	6	Results are inconsistent.		More studies are needed.
19. Precommitment	Kotter, 2019 (19); Ladouceur, 2012 17); Ladouceur, 2017 (5); Livingstone, 2014 (15); McMahon, 2019 (13); Matheson, 2018 (7); Williams, 2012 (11)	47	A conclusive statement on the effectiveness of pre-precommitment cannot yet be offered because of methodological problems, implementation discrepancy and inconsistent results.	It is important to distinguish the target to avoid boomerang effects; universal and binding precommitment systems are effective; a focus on limiting the time spent is priority.	To better design implementation conditions and monitor fidelity.
20. Self-exclusion	Ariyabuddhiphongs, 2013 (10); Drawson, 2017 (14); Gainsbury, 2014 (14); Ladouceur, 2017 (9); Livingstone, 2014 (18); McMahon, 2019 (11); Matheson, 2018 (7); Williams, 2012 (16)	47	A conclusive statement on the effectiveness of self-exclusion cannot yet be offered because of methodological problems, implementation discrepancy and inconsistent results. Self-excluders generally experience benefits from the program.	The reviews analyzed offer several suggestions about the effectiveness of implementation conditions and elements that should be included in self-exclusion programs were suggested.	To better design implementation conditions and monitor fidelity.

N = number of unique studies.

## Data Availability

Data sharing not applicable.

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
