# Peer review of "Prevention and Harm Reduction Interventions for Adult Gambling at the Local Level: An Umbrella Review of Empirical Evidence"

_ijerph, 2021, doi:10.3390/ijerph18189484_

Round 1
Reviewer 1 Report
This is a very well conducted, written, and presented research. Moreover, the topic is important and likely to be of interest to both academics and policy makers.
Minor comments:
- I imagine the authors are using US spelling for this manuscript. However, at few places they have used UK spelling (e.g., behavioural and behaviours).
- On page page 1/line 37 (after the sentence: On the other hand, policies can have an important role in preventing risk gambling, reducing risk factors, and minimizing harms related to gambling), I recommend adding the following sentence:
‘For example, cognitive psychology research reveals that people can benefit from polices that require organizations to provide full utilitarian descriptions regarding the tasks and their consequences; specifically, it is argued that such accessibility of information enhances people’s utilitarian (rational) behavior (Kusev et al., 2016; Martin et al., 2017).
Kusev, P.; van Schaik, P.; Alzahrani, S.; Lonigro, S.; Purser, H. Judging the morality of utilitarian actions: How poor utilitarian accessibility makes judges irrational. Psychon. Bull. Rev. 2016, 23, 1961–1967.
Martin, R.; Kusev, I.; Cooke, A.; Baranova, V.; van Schaik, P.; Kusev, P. Commentary: The social dilemma of autonomous vehicles. Front. Psychol. 2017, 8, 1–2.
- I would also like to bring to the attention of the authors a recently published review article in IJERPH on problem gambling (please see below). I think, the inclusion of this research will enhance further the scope of this excellent manuscript.
Problem Gambling ‘Fuelled on the Fly’ ( https://www.mdpi.com/1660-4601/18/16/8607 )
Author Response
Dear reviewer,
We thank you for your thoughtful suggestions and insights, which have helped us improved the paper significantly. We are also grateful for your positive feed-backs.
The manuscript has been rechecked and the necessary changes have been made in accordance with your suggestions. The responses to all comments have been prepared and given below.
- I imagine the authors are using US spelling for this manuscript. However, at few places they have used UK spelling (e.g., behavioural and behaviours).
Response: Thanks for the comment. We checked the spelling and corrected the words with UK spelling.
- On page page 1/line 37 (after the sentence: On the other hand, policies can have an important role in preventing risk gambling, reducing risk factors, and minimizing harms related to gambling), I recommend adding the following sentence:
‘For example, cognitive psychology research reveals that people can benefit from polices that require organizations to provide full utilitarian descriptions regarding the tasks and their consequences; specifically, it is argued that such accessibility of information enhances people’s utilitarian (rational) behavior (Kusev et al., 2016; Martin et al., 2017).
Kusev, P.; van Schaik, P.; Alzahrani, S.; Lonigro, S.; Purser, H. Judging the morality of utilitarian actions: How poor utilitarian accessibility makes judges irrational. Psychon. Bull. Rev. 2016, 23, 1961–1967.
Martin, R.; Kusev, I.; Cooke, A.; Baranova, V.; van Schaik, P.; Kusev, P. Commentary: The social dilemma of autonomous vehicles. Front. Psychol. 2017, 8, 1–2.
Response: Thanks for your useful suggestion. We included the sentence and the references recommended.
- I would also like to bring to the attention of the authors a recently published review article in IJERPH on problem gambling (please see below). I think, the inclusion of this research will enhance further the scope of this excellent manuscript.
Problem Gambling ‘Fuelled on the Fly’ ( https://www.mdpi.com/1660-4601/18/16/8607 )
Response: Thanks for the very useful suggestion, which has helped us improved the paper significantly. We couldn’t include the paper suggested among the studies selected by the revision because it didn’t respect the criteria of selection. However, the study gave us interesting inputs and we referred to the study results in the introduction.
Reviewer 2 Report
This is an excellent paper dealing with the prevention and harm reduction interventions for adult gambling at the local Level. The main research findings of this paper will be important for the full understanding of the prevention and harm reduction interventions for adult gambling phenomenon. The main analysis is well described and will likely become a cited example of how to undertake such tasks. The prevention and harm reduction interventions for adult gambling described are more straightforward and follow the results of a number of previous studies. This paper is an important contribution and I recommend that it be accepted for publication with minor revision. I suggest that it is necessary to publish here the statistical processing of the meta-analysis in more objective manner. While this paper is important in reviewing previous studies, I believe that the methods of meta-analysis need to be more specifically defined. 
Author Response
Dear reviewer,
We thank you for your thoughtful suggestions and insights, which have helped us improved the paper significantly. We are also grateful for your positive feed-backs.
The manuscript has been rechecked and the necessary changes have been made in accordance with your suggestions. The responses to all comments have been prepared and given below.
- I suggest that it is necessary to publish here the statistical processing of the meta-analysis in more objective manner. While this paper is important in reviewing previous studies, I believe that the methods of meta-analysis need to be more specifically defined. 
Response: Thanks for your important suggestion. However, the study is not a meta-analysis. The umbrella review does not include statistical processing of the results. It synthesizes research evidence and compiles evidence from multiple reviews into one accessible and usable document (Grant & Booth, 2008). We included this issue in the limitations section.
Reviewer 3 Report
This paper presents results of a umbrella review of the effectiveness of gambling preventive and harm reduction strategies, which can be implemented at a local level and targeted at adults. Sixteen reviews are analyzed, and 20 strategies are identified and classified into four areas: supply reduction, demand reduction, risk reduction, and harm reduction.
This is a comprehensive and interesting summary, useful for stakeholders. The methodology is well described and robust. The results are clearly displayed and well discussed.
The results are a little disappointing because they mainly show the gaps in knowledge in these fields due to relatively few conclusive results, linked to a still low number of researches in this field. Obviously, this is not the responsibility of the authors. Also, the main interest of this paper is to indicate, what the authors are doing well, the areas in which it is necessary to work in the future.
This paper looks good to me for publication.
I make a few suggestions which I submit to the authors for consideration.
- Of the 20 strategies outlined and discussed, a majority of them rely on the goodwill of operators regarding their implementation. This is aptly mentioned by the authors. But this issue seems to me of great importance and could be more widely discussed. Indeed, we know that a significant part of the turnover of gambling industry is based on the activity of problem gamblers and this objective fact implies that an gambling operator can only have a limited will to prevent the gambling problem. This is a structural brake on the implementation of these measures. The notion of responsible gambling would also be discussed.
- Faced with the scant evidence in the efficacy or effectiveness of gambling preventive and harm reduction strategies and while waiting for things to improve, we cannot stop there. Older and validated knowledge from other areas of addiction (tobacco, alcohol, drugs) can be mobilized and give us a good indication of what might also work for gambling. I am thinking in particular of the following summary works (non-exhaustive list): Anderson & Baumberg, 2006; Babor, 2003; Edwards, 1995; McNeill et al., 2004; Strang et al., 2012
Finally, I have two minor remarks:
- I did not understand the meaning of the following sentence in the introduction "Regulations have a crucial role in this area, promoting or reducing gambling supply or reducing risks and harms. ”(Lines 31, 32).
- Also in the introduction, it says “Regulation has contributed to enhancing the offer of gambling and to further its reach into everyday life. Therefore, the social acceptability of gambling has changed, and the risks of this behavior are often underestimated ”(lines 33-35), in what sense the social acceptability of gambling has changed? Increased? Decreased?
Author Response
Dear reviewer,
We thank you for your thoughtful suggestions and insights, which have helped us improved the paper significantly. We are also grateful for your positive feed-backs.
The manuscript has been rechecked and the necessary changes have been made in accordance with your suggestions. The responses to all comments have been prepared and given below.
- Of the 20 strategies outlined and discussed, a majority of them rely on the goodwill of operators regarding their implementation. This is aptly mentioned by the authors. But this issue seems to me of great importance and could be more widely discussed. Indeed, we know that a significant part of the turnover of gambling industry is based on the activity of problem gamblers and this objective fact implies that an gambling operator can only have a limited will to prevent the gambling problem. This is a structural brake on the implementation of these measures. The notion of responsible gambling would also be discussed.
Response: Thanks for the very useful suggestion, which has helped us improved the paper significantly. We discussed this issue more in-depth and we better integrated comments related to this issue mentioned in different parts of the paper.
Moreover, we referred to the notion of responsible gambling both in the introduction and in the discussion. This concept is strictly related to the health promotion and “health in all policies” approaches. We also referred to harm reduction in gambling and consumer protection notions because these terms and concepts are often used in the literature (Tanner et al., 2017).
- Faced with the scant evidence in the efficacy or effectiveness of gambling preventive and harm reduction strategies and while waiting for things to improve, we cannot stop there. Older and validated knowledge from other areas of addiction (tobacco, alcohol, drugs) can be mobilized and give us a good indication of what might also work for gambling. I am thinking in particular of the following summary works (non-exhaustive list): Anderson & Baumberg, 2006; Babor, 2003; Edwards, 1995; McNeill et al., 2004; Strang et al., 2012
Response: Thanks for the very useful suggestion, which has helped us improved the paper significantly. We agree that using older and validated knowledge from other areas of addiction to define recommendations for gambling prevention is crucial. Practitioners and policy-makers have used this approach but few studies have considered it. We suggested this approach as a research implication and we underlined that the Gainsbury et al. (2014) review (included in our umbrella review) gave recommendations about gambling policies examining the evidence-based alcohol policies based on the framework provided by Babor et al. (2010).
- I did not understand the meaning of the following sentence in the introduction "Regulations have a crucial role in this area, promoting or reducing gambling supply or reducing risks and harms. ”(Lines 31, 32).
Response: Thanks for the very useful suggestion, which has helped us improved the paper. We clarified the sentence.
- Also in the introduction, it says “Regulation has contributed to enhancing the offer of gambling and to further its reach into everyday life. Therefore, the social acceptability of gambling has changed, and the risks of this behavior are often underestimated” (lines 33-35), in what sense the social acceptability of gambling has changed? Increased? Decreased?
Response: Thanks for the very useful suggestion, which has helped us improved the paper. We clarified that the social acceptability of gambling has increased.